# COVID-19 Vaccination Booster Dose: Knowledge, Practices, and Intention among Pregnant/Planning to Get Pregnant and Lactating Women

**DOI:** 10.3390/vaccines11071249

**Published:** 2023-07-17

**Authors:** Walid Al-Qerem, Anan Jarab, Yara Shawabkeh, Jonathan Ling, Alaa Hammad, Badi’ah Alazab, Fawaz Alasmari

**Affiliations:** 1Department of Pharmacy, AlZaytoonah University of Jordan, Amman 11733, Jordan; 201917008@std-zuj.edu.jo (Y.S.); alaa.hammad@zuj.edu.jo (A.H.); b.alazab@zuj.edu.jo (B.A.); 2College of Pharmacy, Al Ain University, Abu Dhabi P.O. Box 64141, United Arab Emirates; asjarab@just.edu.jo; 3Department of Clinical Pharmacy, Faculty of Pharmacy, Jordan University of Science and Technology, Irbid 22110, Jordan; 4Faculty of Science and Wellbeing, University of Sunderland, Sunderland SR1 3SD, UK; jonathan.ling@sunderland.ac.uk; 5Department of Pharmacology and Toxicology, College of Pharmacy, King Saud University, Riyadh 12372, Saudi Arabia; ffalasmari@ksu.edu.sa

**Keywords:** COVID-19 vaccine, booster dose, intention, pregnant, lactating

## Abstract

Pregnant women are at higher risk of developing severe COVID-19 symptoms. Therefore, booster dose against COVID-19 was recommended for this special population in Jordan. However, vaccine hesitancy/refusal remains the main obstacle to providing immunity against the spread of COVID-19. Thus, the aim of this study is to examine the intention of pregnant/planning to get pregnant and lactating women towards receiving a booster dose against COVID-19 and its associated factors. A questionnaire was given to Jordanian pregnant/planning to get pregnant and lactating females. A total of 695 females were enrolled in the study. Older age, having a chronic disease, high education, high income, and high perceived risk of COVID-19 were significantly associated with higher knowledge about COVID-19. High perceived risk of COVID-19 was significantly associated with better practice. Participants who anticipated they might contract COVID-19 in the next six months, had high perceived risk of COVID-19, had high knowledge, had received the COVID-19 vaccine based on conviction, and smokers had higher intention to receive a booster dose of the COVID-19 vaccination. In order to increase pregnant and lactating women’s intention to receive a booster dose of the COVID-19 vaccine, public health organizations should consider developing comprehensive health education campaigns.

## 1. Introduction

Coronavirus disease 2019 (COVID-19) emerged in December 2019 in Wuhan, China. According to the (World Health Organization) WHO, as of 19 April 2023, there have been over 763 million confirmed cases of the disease and more than six million deaths globally [1]. It is estimated that by the end of the pandemic the total financial cost of COVID-19 will likely be between USD 8.1–15.8 trillion worldwide [2].

Although there are emerging medications to treat COVID-19, preventative vaccination has been an extremely effective measure in reducing the incidence of COVID-19 and its complications, notably serious disease and disease-attributable mortality [3,4]. Nonetheless, vaccine refusal and hesitancy are significant barriers for vaccination programs’ success. Previous studies found that frequent reasons for vaccine hesitancy are personal and religious beliefs, wanting more information from healthcare providers, and safety concerns [5]. Due to new SARS-CoV-2 variants and the reduction of serum anti-spike IgG antibody levels that occur several months after vaccination, booster vaccination strategies have been implemented in several regions and are recommended globally by different health organizations, including the WHO [6,7].

Jordan has implemented a range of comprehensive and stringent preventive measures at multiple levels to effectively combat the transmission of COVID-19. However, as of 25 May 2022, authorities in Jordan have implemented relaxed COVID-19 measures. The compulsory use of facemasks in both indoor and outdoor public areas is no longer mandatory. Furthermore, all restrictions on public gatherings, including those at places of worship and wedding venues, have been lifted, but all the preventable instructions are still recommended including wearing face masks, maintaining a social distance of 3 feet in indoor settings, and hand hygiene. These decisions align with a decline in the number of COVID-19 cases [8,9].

Pregnant women are at higher risk of developing severe COVID-19 symptoms which may lead to more severe complications that require hospitalizations than non-pregnant women. A previous study suggested an association between the severity of COVID-19 during pregnancy and the Delta (B.1.617.2) variant of the SARS-CoV-2 virus [10]. Accordingly, The Centers for Disease Control and Prevention (CDC) recommend COVID-19 booster vaccines for pregnant females and consider them as a high-risk group. Similarly, CDC classifies lactating woman as a high-risk group receive COVID-19 vaccine [11].

Previous studies have shown suboptimal vaccination acceptance rates among pregnant and lactating women. For example, an Iranian study reported that only 42.9% of pregnant women consented to receive the COVID-19 vaccine [12]. A high rate of vaccine hesitancy was reported in a study from Italy (86.4%). This study revealed that the main reported reasons for hesitancy were concerns about the safety of the vaccine, no recommendation from a gynecologist, and a lack of knowledge [13]. Similarly, several studies have reported high hesitancy among lactating women in receiving the COVID-19 vaccine [14,15]. High hesitancy/refusal rates for receiving the COVID-19 vaccine in this subgroup could be attributed to the limited clinical studies that have enrolled pregnant women [16].

As far as we know, no previous studies have assessed Jordanian lactating, pregnant, or planning to get pregnant females’ intention to receive a booster dose against COVID-19. Thus, the primary aim of the present study is to evaluate the intention of these special populations toward receiving the first booster dose against COVID-19 (which will be referred to in the rest of the manuscript as the booster dose) and its associated factors, the study will also evaluate participants’ knowledge and practices towards COVID-19 and its vaccine.

## 2. Materials and Methods

This study is a multi-center cross-sectional study that was conducted across six different gynecologist clinics in Jordan. The distribution of participants is as follows: two clinics in Amman, with a combined total of 290 participants; one clinic in Irbid, which enrolled 150 participants; one clinic in Zarqa, which enrolled 140 participants; one clinic in Madaba, which recruited 138 participants; and one clinic in Aqaba, with a total of 132 participants. The current study included only the pregnant/planning to get pregnant and lactating females who were previously vaccinated with two doses of the COVID-19 vaccine. The research pharmacist approached pregnant/planning to get pregnant and lactating women attending different gynecologist clinics in Jordan. A quick interview was conducted with each participating woman to ensure that she met the inclusion criteria. The research pharmacist provided the enrolled women with an information sheet describing the study details and purposes. The researcher emphasized that participation was voluntary, and the participants had the right to refuse or withdraw from the study at any time without any effect on the provided healthcare. Women who agreed to participate were asked to sign a consent form. A Google Forms link was sent to the enrolled participants to complete the study questionnaire. Google Forms eliminates any possibility of missing data as it does not allow the participants to submit the form before completing all the required questions. The data were collected between May 2022 and April 2023. The principles of the Declaration of Helsinki were met, and the ethical approval was conceded by the Al-Zaytoonah Research Ethics Committee.

### 2.1. Sampling Type and Sample Size

Convenience sampling was adopted for participant recruitment in the present study. To calculate the minimum required sample size, the Krejcie and Morgan formula [17] was applied.

The formula is:s = X^2^NP(1 − P) + d^2^ (N − 1) + X^2^P(1 − P)

s = required sample size;

X^2^ = the table value of chi-square for 1 degree of freedom at the desired confidence level (3.841);

N = the population size;

P = the population proportion;

d = the degree of accuracy expressed as a proportion (0.5).

Krejcie and Morgan composed sample size tables based on the above formula for different population sizes, confidence levels, and margin of errors. The population size was assumed to be above 1,000,000 (indefinite) and the minimum required sample was 384 for 95% CI and 5% margin of error.

### 2.2. Study Instrument

The present study’s questionnaire is based on a previously published study by Al-Qerem et al. [18] that evaluated intentions toward receiving the COVID-19 booster dose among the general population after adapting it to pregnant/planning to get pregnant and lactating females by adding relevant questions to these specific populations. These questions were added after an extensive literature review that evaluated intentions toward receiving the COVID-19 vaccine among the designated population [19,20,21].

The questionnaire was composed of several sections. The first section gathered participants’ sociodemographic characteristics. The second section evaluated the participants’ previous experiences and attitudes toward COVID-19 and included questions about previous infection, perceived possibility of future reinfection with COVID-19, knowing someone who died due to COVID-19, and the perceived risk of COVID-19 infection, which was evaluated by a single question with a response scale from 1 (indicating very low perceived risk) to 5 (indicating very high risk). The next part evaluated participants’ experiences and attitudes toward the COVID-19 vaccine, which included reasons for previous vaccination against COVID-19, type of vaccine received, intention to receive a booster dose, and side effects experienced due to the COVID-19 vaccine. A side effects score was computed by adding up all the reported side effects with a maximum possible score of 10. The fourth section evaluated participants’ knowledge about COVID-19 symptoms, protective practices against COVID-19, methods of infection transmission, and management of COVID-19, which included 22 items; one point was granted for each correct answer, while no points were granted for wrong answers, and then the sum of the points of the correct answers was used to compute the knowledge scale with a maximum possible score of 22. The fifth part evaluated participants’ adherence to protective practices against COVID-19, which included 5 items: wearing face masks, washing hands with soap regularly, using detergent, social distancing, and avoiding touching face/mouth/nose/eyes. The response scale for these items ranged from 1 point for “never” to 5 points for “always”, with the highest possible score of 25. The final part evaluated reasons for not receiving the booster dose among those who were reluctant/refused to receive the booster dose. The questionnaire is available in the Appendix A.

### 2.3. Survey Validity and Reliability

The questionnaire was initially written in English and then translated into Arabic before it was back translated to English by native English speakers. The two English versions were deemed comparable. The content validity was verified by a panel of experts that included public health specialists, clinical pharmacists, and medical doctors specializing in infectious diseases. To determine face validity, the questionnaire was given to 40 participants and their feedback was obtained to enhance the questionnaire items. The data collected from the pilot study was excluded from the final dataset. The present study composed two latent variables: knowledge about COVID-19 and the adherence of protective practices against COVID-19; the reliability of these two latent variables was evaluated by examining Cronbach’s alpha value to ensure internal consistency.

### 2.4. Statistical Analysis

All statistical analyses were conducted using SPSS version 28. Categorical variables were presented as frequencies and percentages. The normality of the continues variables, perceived risk, side effects score, knowledge score, and practice score were assessed using Q-Q plots, and these plots indicated that all the continuous variables were not normally distributed; therefore, continuous variables were presented as median 95% CI and nonparametric analysis was conducted. The medians for the knowledge and practice scores were computed and the participants who were above the median were included in the “High” level group, while the least were included in the “Low” level group and the median for each score was reported in the result section. To identify variable associations with knowledge and practice, bivariate analyses using Chi-square and Mann–Whitney U tests were performed. For intention to receive a booster dose of COVID-19, Chi-square and Kruskal–Wallis tests were applied. Two binary regression models were built to assess variables associated with knowledge and practice levels, and a multinomial logistic regression model was performed to evaluate variable association with the intention to take a booster dose of COVID-19. The independent variables in the three regression models were defined as the variables that had *p*-values less than 0.2 in the bivariate analysis. A significant level was determined at *p*-value < 0.05.

## 3. Results

In total, 850 females who met the inclusion criteria were invited to participate in the present study, of whom 695 females (82%) agreed to participate. Sample demographic characteristics show that over half of the sample were between 18 and 29 years old (62.7%). Half of the enrolled participants were planning to get pregnant (51.1%), and almost three-quarters held a bachelor’s degree or higher (72.4%) (Table 1).

The normality of the continuous variables, perceived risk, side effect score, knowledge score, and practice score were assessed using Q-Q plots. These plots indicated that the all of the continuous variables were not normally distributed; therefore, nonparametric analyses were conducted. Most of the sample (77.7%) reported knowing someone who died due to COVID-19, and 50.4% of them reported previously being infected with COVID-19, while 18.6% were unsure if they had been infected (Table 2).

Participants’ experiences and attitudes toward the COVID-19 vaccine are shown in Table 3. The most frequently received vaccination was Pfizer’s (74.8%), and the most reported reason for taking the first doses of COVID-19 vaccine was conviction (67.8%). More than half of the enrolled participants intended to receive the booster dose of the COVID-19 vaccine (59.1%), while 23.5% of them were unsure. The most stated level of side effects due to COVID-19 vaccine severity was moderate (41%), and the most reported side effects were pain at the site of injection and fever (68.1% and 56.4%, respectively). The least reported side effect was water retention (1.2%). The median for the side effects score was 3 (3–4) out of 10.

As shown in Table 4, the most frequent knowledge item about the symptoms of COVID-19 was fever (95.4%), followed by loss of smell and taste (92.4%), while the least frequent was otitis media (34%). Additionally, the highest correct answer for protective practices were social distancing and wearing face masks (94.5% and 91.8%, respectively), while the lowest was medical herb consumption (44%). Furthermore, knowledge about the transmission of COVID-19 was obtained, and the most frequently correct item was the inhalation of respiratory droplets from an infected person (96.4%), while the least was eating or touching wild animals (42.2%). Regarding knowledge about the management of COVID-19, the most frequently reported correct item was for “What is the protective time that the vaccine provides against COVID-19?” (68.5%), while the least was “Is it necessary for the booster dose of the vaccine to be of the same type of vaccine used in the initial doses?” (27.9%).

The median for the Knowledge scores was 15 (15–16) out of a maximum possible score of 22. Knowledge scores were divided into two categories based on each score median, with those who scored less than or equal to the median classified as “low”, while the rest were classified as the high-knowledge group. The internal consistency of the knowledge scale was confirmed by computing Cronbach’s alpha (0.714) (Table 4).

Participants’ responses to the practice items are included in Table 5. The highest always/usually performed practices were washing hands with regular soap, followed by wearing face masks (91.6% and 82.4%, respectively), while the lowest was avoiding touching face/mouth/nose/eyes (70.5%). The median for the practice score was 22 (22–23) out of a maximum possible score of 25. The scores were divided into two categories based on each score median. Cronbach’s alpha was computed, and the results revealed that the practice scale had high internal consistency (0.841).

Bivariate analysis using Chi-square and Mann–Whitney U tests assessed the association between the knowledge and practice levels. Age was significantly associated with knowledge level (*p* < 0.001). Along with pregnancy status (*p* < 0.001), education (*p*-value < 0.001), household monthly income level (*p* < 0.001), chronic diseases (*p* = 0.12), perceived possibility of COVID-19 infection in the next 6 months (*p* < 0.001), and how serious the participants thought the infection with COVID-9 was (*p* < 0.001) were all associated with knowledge. Furthermore, the variables that were significantly associated with practice level were how serious the participants thought infection with COVID was (*p* < 0.001) and education (*p* = 0.09) (Appendix A).

The variables associated with intention to get a booster dose of COVID-19 are presented in Appendix A. The results show that the vaccine type (*p*-value = 0.002), age (*p*-value = 0.016), pregnancy status (*p* = 0.006), education (*p* = 0.011), household monthly income (*p* = 0.003), smoking (*p* < 0.001), reason to receive COVID-19 vaccine (*p* < 0.001), reporting knowing someone who died due to COVID-19 (*p* = 0.029), perceived possibility to be infected with COVID-19 in the next 6 months (*p* < 0.001), perceived risk the infection with COVID-19 is (*p* < 0.001), knowledge level (*p* < 0.001), and finally practice level (*p* < 0.001) were significantly associated with the intent to receive COVID-19.

Table 6 shows the binary regression results for variable association with the knowledge level. Participants aged from 30 to 39 had higher odds of being in the high-knowledge group compared to participants aged 18–29 years old (OR = 1.576, 95% CI (1.071–2.319), *p* = 0.021). Participants with chronic diseases had higher odds of being in the high-knowledge group compared to those with no chronic diseases (OR = 1.913, 95% CI (1.007–3.632), *p* = 0.047). Likewise, those who were highly educated had higher odds of being in the high-knowledge group compared to those with lower education (OR = 2.959, 95% CI (1.896–4.618), *p* < 0.001). Furthermore, having household income between JOD 500 and 1000 and more than JOD 1000 increased the likelihood of being in the high-knowledge group compared to those with a household income less than JOD 500 (OR = 2.713, 95% CI (1.803–4.083), *p* < 0.001; and OR = 4.803, 95% CI. (2.938–7.852), *p* < 0.001, respectively). Finally, as the perceived risk of COVID-19 increased, the odds of being in the high-knowledge group increased (OR = 1.361, 95% CI (1.169–1.586), *p* < 0.001).

The binary regression model for variables associated with the practice level is displayed in Table 7. “How serious is the infection with COVID?” was the only significant variable that increased the odds of being in the high-practice group (OR = 1.272, 95% CI (1.116–1.450), *p* < 0.001).

Multinomial regression was conducted to assess the variables related to responding “No” or “Maybe” vs. “Yes” regarding the intention to receive a booster dose of the COVID-19 vaccine. Participants who believed they would not be infected with COVID-19 in the next six months had higher odds of answering “No” when compared to those who believed they would be infected with severe symptoms (OR = 8.667, 95% CI (1.569–47.881), *p* = 0.013). Furthermore, the low-knowledge group had higher odds of answering “No” or “Maybe” when compared to the high-knowledge group (OR = 1.875, 95% CI (1.062–3.311), *p* = 0.030). Participants whose reason for receiving the COVID-19 vaccine was imposed laws or both conviction and imposed laws had higher odds of answering “No” (OR = 19.859, 95% CI (10.059–39.207), *p* < 0.001; and OR = 2.604, 95% CI. (1.344–5.043), *p* = 0.005, respectively) or “Maybe” (OR = 4.369, 95% CI = 2.241, *p* = 8.518; and OR = 2.081, 95% CI (1.221–3.546), *p* < 0.001, respectively) when compared to conviction only. Non-smoking participants had higher odds of answering “Maybe” when compared to smokers (OR = 3.774, 95% CI (1.916–7.432), *p* < 0.001). Finally, as the perceived risk of COVID-19 increased, the likelihood of answering “No” decreased (OR = 0.750, 95% CI (0.604–0.931), *p* = 0.009) s(Table 8).

Those who answered “No” to the question “Do you intend to receive the booster dose for COVID-19 vaccine?” were assigned to the vaccine refusal group, while participants who answered “Maybe” were assigned to the vaccine hesitancy group. The main reasons for reported vaccine hesitancy/refusal included “Lack of sufficient studies that evaluated the safety of COVID-19 on pregnant/lactating women” which was reported by 94% of the participants who refused or were reluctant to receive the booster dose. This was followed by “I am worried that the booster dose will harm my fetus/infant” (90.8%), while the least reported reason was “I was infected with COVID-19, therefore I do not need the booster dose” (Appendix A).

## 4. Discussion

The majority of participants intended to receive the booster dose of the COVID-19 vaccine (59.1%). In comparison, nearly half of the participants surveyed in a study conducted in Ghana showed their acceptance of the COVID-19 vaccine booster dose [22], while higher acceptance of a booster dose of the COVID-19 vaccine was reported in studies conducted in Australia [23], Pakistan [24], and Malaysia [25].

In the current study, the majority of participants indicated that they encountered pain at the injection site (68.1%) and experienced fever (56.4%) following administration of the COVID-19 vaccine. The Centers for Disease Control and Prevention (CDC) have also highlighted that in terms of COVID-19 vaccine safety, the majority of individuals have reported mild-to-moderate side effects like headaches, fatigue, and soreness at the injection site, which typically subside within a few days, whereas fever and chills are commonly experienced by adults aged 18 years or older [26,27]. Findings from a systematic review of evidence on the safety data from COVID-19 vaccine trials indicated that the most frequently reported local adverse events were pain, swelling, and redness at the injection site, while systemic reactions included fever, fatigue, muscle pain, and headaches [28].

Although the majority of the participants in the present study expressed good knowledge regarding COVID-19, there were certain gaps in knowledge with regard to COVID-19 symptoms, protective practices, methods of transmission, and management. Specifically, a significant number of participants were unaware that otitis media can be a symptom of COVID-19 (66%), did not recognize the protective effects of medical herb consumption against COVID-19 (55.7%), were unaware of the potential transmission risks associated with eating or touching wild animals (57.8%), and lacked knowledge about the booster dose of the vaccine (72.1%). Therefore, addressing these knowledge gaps and misconceptions through targeted education, awareness campaigns, and accurate information dissemination can contribute to better public health outcomes, improved prevention practices, and effective management of future infectious disease outbreaks.

Based on the observations from the current study, it is encouraging to see that participants generally displayed good adherence to protective precautions against COVID-19. The result indicated high protective practices, including adherence to wearing masks and preserving social distancing; this is similar to a previous Saudi study that reported high levels of awareness in relation to social distancing as a preventive step to control the spread of COVID-19 [29,30,31]. Therefore, healthcare authorities should exert efforts to educate the public about the risks associated with close contact and face-touching, as well as provide information on alternative ways to greet, interact, and manage habits. People can better appreciate the importance of these measures and be encouraged to modify their behavior through educational campaigns.

In the current study, several sociodemographic and medical characteristics, such as older age, having a chronic disease, and high education, were significantly associated with higher knowledge about COVID-19, which may be justified by the superior awareness of this population with regard to the COVID-19 pandemic and its associated risks. Higher income was also linked to better knowledge in the present study. The reason behind this finding is likely that individuals with higher income tend to have greater access to diverse information sources, such as trustworthy news platforms and subscriptions to scientific journals. This enhanced accessibility to information sources can expose them to more precise and current information about COVID-19, ultimately resulting in a higher level of knowledge. A study conducted in Brazil found that participants who had a higher education level had better knowledge about COVID-19 than their peers [32]. A systematic review reported that the education, income, and age were among the significant factors associated with poor knowledge regarding the pandemic of COVID-19 [33]. Another study found that Ghanaians’ knowledge about COVID-19 was significantly affected by their age and education [34]. Ethiopian healthcare professionals’ knowledge about COVID-19 was also affected by their age, education, and income [35]. Furthermore, the findings of our study showed a significant relationship between perceived risk of COVID-19 and both knowledge about it and practice of the preventive measures against it. This discovery might be connected to a greater concern exhibited about the pandemic, which increases their curiosity about the disease and their keenness to practice the preventive precautions to avoid getting infected with COVID-19. According to a qualitative study, perceptions of the risk of COVID-19 are impacted by information and guidance campaigns, which in turn shapes their protective behavior [36]. Based on these study results, it is important to develop educational campaigns specifically tailored to reach populations with lower levels of knowledge about COVID-19, with particular focus on younger individuals, those with lower education, and lower incomes.

If participants in the current investigation anticipated that they might contract COVID-19 in the next six months, their intention to receive a booster dose of the COVID-19 vaccination significantly increased. Perception of the risk connected to COVID-19 can be increased if people anticipate the likelihood of contracting it soon. Individuals may seek out additional precautions, such as booster vaccine doses, to reduce the probability of infection as a result of their elevated feeling of risk. This was also observed in our study, where participants expressed a much greater intention to receive a booster dose of the COVID-19 vaccination when they believed the disease to be more threatening. Consistent results were reported in earlier research [23,37]. Furthermore, participants who had high knowledge about COVID-19, and those who received the COVID-19 vaccine based on conviction, had significantly higher intention to take a booster dose of the vaccine. Participants in Jordan who were legally forced to obtain COVID-19 vaccination, as opposed to those who received the vaccination out of personal conviction, showed greater resistance to receiving a third, booster, shot [18]. These results imply that education about COVID-19 and the necessity of the booster dose, rather than mandating vaccination through the legal system, may raise people’s intention to get the vaccine. This can be utilized as a productive strategy to encourage the uptake of the booster dose of the COVID-19 vaccine. Smokers in our study showed higher intention to take the vaccine than nonsmokers. This association may be explained by smokers’ greater concern about the potentially serious symptoms this virus may have on their lungs when compared to nonsmokers [38,39], which will enhance their desire to receive the booster dose of the vaccine to lower their risk of contracting the infection.

The most frequently reported reasons for vaccine hesitancy/refusal in the present study were concerns about the harmful effects of the booster dose of COVID-19 on the developing fetus, and lack of sufficient studies that evaluated the safety of COVID-19 on pregnant/lactating women. Similar results were reported in studies conducted in Iran [40], Italy [13], the UK [41], and Saudi Arabia [15], where most of the participating pregnant women reported that the reason behind their refusal of COVID-19 vaccination was their fear of its side effects on the fetus. According to a study conducted in the UK, a lack of research about long-term outcomes for the baby was among the most common reasons for COVID-19 vaccine uptake refusal in pregnancy [42]. This lack of direct evidence on the safety and efficacy of COVID-19 vaccines in pregnant and lactating populations has contributed to hesitancy among individuals in these groups. Therefore, strong research studies that especially assess the safety and effectiveness of COVID-19 vaccinations in pregnant and breastfeeding women must be conducted. By filling in these knowledge gaps, comprehensive research can reduce concerns and give precise data to inform vaccination decisions in these communities.

### Study Strengths and Limitations

This is a multi-centered study which may increase the reliability and generalizability of the study results. Moreover, the present study did not only evaluate intention to receive a COVID-19 booster dose but also evaluated its association with different factors, including previous experience with COVID-19 and COVID-19 vaccines, knowledge about the COVID-19 vaccine, and adherence to protective practices, in addition to different demographic variables. However, the cross-sectional design used in this study limits its ability to establish a cause–effect relationship. The convenience sampling technique could have led to selection bias. Additionally, the study relied on self-reported data, which may be subject to social desirability and recall biases.

## 5. Conclusions

In the present study, a sample of pregnant/planning to get pregnant and lactating women were asked about their intention to receive the COVID-19 booster dose. As the results showed, a substantial portion of this special population were hesitant or refused to receive the COVID-19 booster dose. Low knowledge about COVID-19 was one of the main variables associated with booster dose refusal. This is also evident in the participants who had low perceived risk of COVID-19 or assumed that future reinfection with COVID-19 was unlikely. The participants demonstrated average knowledge about COVID-19 and its vaccine, and the main variables that were associated with knowledge included age, chronic disease status, and socioeconomic status. Lastly, the participants reported high adherence toward protective practices against COVID-19, particularly among participants who had high perceived risk of COVID-19.

### Practice Implications

Public health organizations should consider developing comprehensive health education campaigns to ensure accurate knowledge dissemination, improve awareness about COVID-19 vaccination, and increase the intention of pregnant and lactating women to receive a booster dose of the vaccine. In addition, comprehensive research is required to give precise data on the safety and effectiveness of COVID-19 vaccination in pregnancy and breastfeeding to reduce these women’s concerns regarding this issue.

## Figures and Tables

**Table 1 vaccines-11-01249-t001:** Sociodemographic characteristics of participants.

		Frequency (%)
Age	18–29 years old	436 (62.7%)
	30–39 years old	259 (37.3%)
Educational level	Low	192 (27.6%)
	High	503 (72.4%)
Household monthly income level	Less than 500 JOD	242 (34.8%)
	500–1000 JOD	300 (43.2%)
	More than 1000 JOD	153 (22%)
Pregnancy status	Planning to get pregnant	355 (51.1%)
	Lactating	138 (19.9%)
	Pregnant	202 (29.1%)
Smoker	No	577 (83%)
	Yes	118 (17%)
Do you have any chronic diseases?	No	638 (91.8%)
	Yes	57 (8.2%)

**Table 2 vaccines-11-01249-t002:** Participants’ experiences and attitudes toward COVID-19.

		Frequency (%), or Median (95% CI.)
Participants’ experiences toward COVID-19.		
Previously infected with COVID-19	No	216 (31.1%)
	Not sure	129 (18.6%)
	Yes	350 (50.4%)
Do you personally know someone who died due to COVID-19	No	154 (22.3%)
	Yes	538 (77.7%)
Participants’ attitudes toward COVID-19.		
Perceived possibility to be infected with COVID-19 in the next 6 months	I don’t think I will be infected	249 (35.8%)
	I think I will be infected with mild symptoms	419 (60.3%)
	I think I will be infected with severe symptoms	27 (3.9%)
	Yes	538 (77.7%)
In your opinion, how serious is the infection with COVID?		3 (3–4)

**Table 3 vaccines-11-01249-t003:** Participants’ experiences and attitudes toward COVID-19 vaccine.

		Frequency (%), or Median (95% CI.)
Do you intend to receive the booster dose for COVID-19 vaccine?	No	121 (17.4%)
	Maybe	163 (23.5%)
	Yes	411 (59.1%)
Participants’ experiences toward COVID-19 vaccine.		
Reasons for receiving COVID-19 vaccines	Imposed laws	110 (16%)
	Conviction and imposed laws	112 (16.3%)
	Conviction	467 (67.8%)
Previously received COVID-19 vaccine	Pfizer	520 (74.8%)
	Astra Zeneca	77 (11.1%)
	Sinopharm	98 (14.1%)
Perceived side effect level due to COVID-19 vaccine	No symptoms	83 (11.9%)
	Mild	232 (33.4%)
	Moderate	285 (41%)
	Severe	95 (13.7%)
Side effects reported by the participant due to COVID-19 vaccine		
Fever	No	303 (43.6%)
	Yes	392 (56.4%)
Headache	No	369 (53.1%)
	Yes	326 (46.9%)
Nausea	No	589 (84.7%)
	Yes	106 (15.3%)
Pain at the site of injection	No	222 (31.9%)
	Yes	473 (68.1%)
Spasm	No	332 (47.8%)
	Yes	363 (52.2%)
Chills	No	614 (88.3%)
	Yes	81 (11.7%)
Muscle pain	No	478 (68.8%)
	Yes	217 (31.2%)
Rash	No	669 (96.3%)
	Yes	26 (3.7%)
Water retention	No	687 (98.8%)
	Yes	8 (1.2%)
Weakness	No	372 (53.5%)
	Yes	323 (46.5%)
No symptoms	Yes	66 (9.5%)
	No	629 (90.5%)
Side effects score		3 (3–4)

**Table 4 vaccines-11-01249-t004:** Participants’ responses to the knowledge items.

		Frequency (%)
Knowledge about the symptoms of COVID-19		
Fever	Yes *	663 (95.4%)
	No	32 (4.6%)
Chills	Yes *	298 (42.9%)
	No	397 (57.1%)
Diarrhea	Yes *	447 (64.3%)
	No	248 (35.7%)
Cough	Yes *	621 (89.4%)
	No	74 (10.6%)
Otitis media (earache)	Yes	459 (66%)
	No *	236 (34%)
Loss of smell and taste	Yes *	642 (92.4%)
	No	53 (7.6%)
No symptoms	Yes *	504 (72.5%)
	No	191 (27.5%)
Knowledge about the protective practices against COVID-19		
Wearing face masks	Yes *	638 (91.8%)
	No	57 (8.2%)
Washing hands with regular soap	Yes *	584 (84%)
	No	111 (16%)
Using detergents	Yes *	627 (90.2%)
	No	68 (9.8%)
Social distancing	Yes *	657 (94.5%)
	No	38 (5.5%)
Avoiding touching face/mouth/nose/eyes	Yes *	615 (88.5%)
	No	80 (11.5%)
Avoid consuming meat	Yes	251 (36.1%)
	No *	444 (63.9%)
Medical herb consumption	Yes	387 (55.7%)
	No *	308 (44.3%)
Knowledge about the transmission of COVID-19		
Drinking unclean water	Yes	324 (46.6%)
	No *	371 (53.4%)
Eating unclean food	Yes	362 (52.1%)
	No *	333 (47.9%)
Inhalation of respiratory droplets from an infected person	Yes *	670 (96.4%)
	No	25 (3.6%)
Eating or touching wild animals	Yes	402 (57.8%)
	No *	293 (42.2%)
Knowledge about the management of COVID-19		
Is there currently a drug in pharmacies or hospitals that treats COVID-19 completely?	Yes	270 (38.8%)
	No *	425 (61.2%)
What is the protective time that the vaccine provides against COVID-19?	Days	69 (9.9%)
	Months *	476 (68.5%)
	Years	150 (21.6%)
Is it necessary for the booster dose of the vaccine to be of the same type of vaccine used in the initial doses?	Yes	501 (72.1%)
	No *	194 (27.9%)
	Children	105 (15.1%)
Which people need the booster dose?	Elderly	202 (29.1%)
	All people *	388 (55.8%)

* Indicates the correct answer.

**Table 5 vaccines-11-01249-t005:** Participants’ responses to the practice items.

	Never	Rarely	Sometimes	Usually	Always
	Frequency (%)	Frequency (%)	Frequency (%)	Frequency (%)	Frequency (%)
Wearing face masks	4 (0.6%)	35 (5%)	83 (11.9%)	169 (24.3%)	404 (58.1%)
Washing hands with regular soap	4 (0.6%)	11 (1.6%)	43 (6.2%)	144 (20.7%)	493 (70.9%)
Using detergent	5 (0.7%)	36 (5.2%)	104 (15%)	144 (20.7%)	406 (58.4%)
Social distancing	16 (2.3%)	53 (7.6%)	131 (18.8%)	161 (23.2%)	334 (48.1%)
Avoiding touching face/mouth/nose/eyes	18 (2.6%)	72 (10.4%)	115 (16.5%)	149 (21.4%)	341 (49.1%)

**Table 6 vaccines-11-01249-t006:** Binary regression for variable association with the knowledge level.

		OR	95% CI for OR		*p*-Value
			Lower	Upper	
Age	30–39	1.576	1.071	2.319	0.021
	18–29	(REF)	(REF)	(REF)	(REF)
Status	Lactating	0.644	0.395	1.051	0.078
	Pregnant	0.727	0.486	1.087	0.120
	Planning to get pregnant	(REF)	(REF)	(REF)	(REF)
Education	High	2.959	1.896	4.618	<0.001
	Low	(REF)	(REF)	(REF)	(REF)
Household monthly income	500–1000 JOD	2.713	1.803	4.083	<0.001
	>1000 JOD	4.803	2.938	7.852	<0.001
	<500 JOD	(REF)	(REF)	(REF)	(REF)
Do you have any chronic diseases?	Yes	1.913	1.007	3.632	0.047
	No	(REF)	(REF)	(REF)	(REF)
In your opinion, how serious is the infection with COVID?		1.361	1.169	1.586	<0.001
Previously infected with COVID-19	Not sure	0.858	0.506	1.455	0.570
	Yes	0.890	0.593	1.338	0.576
	No	(REF)	(REF)	(REF)	(REF)
Perceived possibility to be infected with COVID-19 in the next 6 months	I think I will be infected with mild symptoms	1.240	0.850	1.808	0.264
	I think I will be infected with severe symptoms	0.496	0.185	1.333	0.164
	I don’t think I will be infected	(REF)	(REF)	(REF)	(REF)

**Table 7 vaccines-11-01249-t007:** Binary regression for variables associated with the practice level.

		OR	95% CI for OR		*p*-Value
			Lower	Upper	
In your opinion, how serious is the COVID infection?		1.272	1.116	1.450	<0.001
Education	High	1.342	0.953	1.919	0.092
	Low	(REF)	(REF)	(REF)	(REF)
Previously infected with COVID-19	Not sure	0.732	0.470	1.140	0.167
	Yes	0.708	0.501	1.002	0.051
	No	(REF)	(REF)	(REF)	(REF)

**Table 8 vaccines-11-01249-t008:** Multinomial regression model for variable association with the intention to receive a booster dose of COVID-19 vaccine.

		No vs. Yes				Maybe vs. Yes			
		OR	95% IR for OR		*p*-Value	OR	95% CI for OR		*p*-Value
			Lower Bound	Upper Bound			Lower Bound	Upper Bound	
In your opinion, how serious is the COVID infection?		0.750	0.604	0.931	0.009	0.953	0.790	1.150	0.619
Age	18–29	1.339	0.752	2.384	0.321	1.150	0.717	1.842	0.562
	30–39	(REF)	(REF)	(REF)	(REF)	(REF)	(REF)	(REF)	(REF)
Status	Planning to get pregnant	0.952	0.525	1.726	0.871	0.998	0.612	1.627	0.994
	Lactating	1.582	0.753	3.324	0.226	1.540	0.826	2.870	0.174
	Pregnant	(REF)	(REF)	(REF)	(REF)	(REF)	(REF)	(REF)	(REF)
Education	Low	0.602	0.325	1.118	0.108	1.129	0.688	1.852	0.632
	High	(REF)	(REF)	(REF)	(REF)	(REF)	(REF)	(REF)	(REF)
Household monthly income	<500 JOD	1.144	0.533	2.455	0.730	0.857	0.458	1.604	0.629
	500–1000 JOD	1.395	0.684	2.845	0.359	1.423	0.814	2.488	0.216
	>1000 JOD	(REF)	(REF)	(REF)	(REF)	(REF)	(REF)	(REF)	(REF)
Smoker	No	1.904	0.922	3.931	0.082	3.774	1.916	7.432	<0.001
	Yes	(REF)	(REF)	(REF)	(REF)	(REF)	(REF)	(REF)	(REF)
Do you personally know someone who died due to COVID-19	No	1.399	0.796	2.458	0.243	0.668	0.395	1.127	0.130
	Yes	(REF)	(REF)	(REF)	(REF)	(REF)	(REF)	(REF)	(REF)
Reasons for receiving COVID-19 vaccines	Imposed laws	19.859	10.059	39.207	<0.001	4.369	2.241	8.518	<0.001
	Conviction and imposed laws	2.604	1.344	5.043	0.005	2.081	1.221	3.546	0.007
	Conviction	(REF)	(REF)	(REF)	(REF)	(REF)	(REF)	(REF)	(REF)
Vaccine type	Pfizer	0.827	0.417	1.640	0.587	0.952	0.531	1.709	0.870
	Astra Zeneca	0.321	0.095	1.082	0.067	0.543	0.217	1.359	0.192
	Sinopharm	(REF)	(REF)	(REF)	(REF)	(REF)	(REF)	(REF)	(REF)
Side effect level	No symptoms	0.830	0.310	2.223	0.711	1.456	0.587	3.609	0.417
	Mild	0.989	0.446	2.190	0.978	1.710	0.830	3.520	0.145
	Moderate	0.783	0.362	1.697	0.536	1.551	0.772	3.117	0.217
	Severe	(REF)	(REF)	(REF)	(REF)	(REF)	(REF)	(REF)	(REF)
Previously infected with COVID-19	No	0.991	0.558	1.759	0.974	0.855	0.522	1.401	0.535
	Not sure	0.634	0.307	1.307	0.217	1.183	0.691	2.023	0.541
	Yes	(REF)	(REF)	(REF)	(REF)	(REF)	(REF)	(REF)	(REF)
Perceived possibility to be infected with COVID-19 in the next 6 months	I don’t think I will be infected	8.667	1.569	47.881	0.013	1.762	0.614	5.055	0.292
	I think I will be infected with mild symptoms	4.636	0.853	25.198	0.076	1.016	0.366	2.822	0.976
	I think I will be infected with severe symptoms	(REF)	(REF)	(REF)	(REF)	(REF)	(REF)	(REF)	(REF)
Knowledge level	Low	1.875	1.062	3.311	0.030	1.655	1.045	2.622	0.032
	High	(REF)	(REF)	(REF)	(REF)	(REF)	(REF)	(REF)	(REF)
Practice level	Low	1.205	0.726	2.001	0.471	2.161	1.422	3.284	<0.001
	High	(REF)	(REF)	(REF)	(REF)	(REF)	(REF)	(REF)	(REF)

## Data Availability

Available upon request.

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
