# Peer review of "COVID-19 Vaccination Booster Dose: Knowledge, Practices, and Intention among Pregnant/Planning to Get Pregnant and Lactating Women"

_vaccines, 2023, doi:10.3390/vaccines11071249_

Round 1

Reviewer 1 Report

Please see an attached file. 

The quality of English is fine, but the writing style needs improvement in terms of clarity and conciseness.

Reviewer 2 Report

This is an interesting article, that provides new information regarding vaccine hesitancy among pregnant & lactating women in Jordan. I have some small comments to the authors.

Abstract: regarding the background, the description corresponds to the aims or objectives of the study, rather than the background information. More information is needed regarding vaccination rate in pregnant women, or why they are at specific risks if vaccinated during pregnancy. Also, it is important to know if at the moment of the study, pregnant women were or not getting the covid vaccine in Jordan.

With respect to this comment in the abstract: “High perceived dangerousness of COVID-19 was significantly associated with better practice”, the better practice was actually getting the vaccine or being willing to receive it? It is not the same. Please clarify.

Key words: I suggest to review if these words are all considered within the MESH Subheadings (ie. Practice and intention probably are not).

Introduction: It will be important to give information regarding the lack of clinical studies of COVID-19 vaccines that included pregnant and lactating women; most probably the lack of adequate clinical trials for these group of women contributed to vaccine hesitancy among this population and is an information that is relevant to consider. Likewise, it is important to have information regarding the Jordan vaccination program for COVID-19 (it included or not pregnant women).

Minor edits:

A total of 695 females enrolled in the study: were enrolled in the study (abstract)

and safety concerns (5) Due (line 44), end of the sentence is missing.

with the majority had no chronic diseases, (line 124), please review the syntaxis.

As shown in Table 4. The most (line 146); there is no need to end the phrase between Table 4 and “the most”…

perceived dangerousness the infection with COVID-19 is (p-value <0.001) (line 187), please review the syntaxis.

Author Response

 The authors would like to thank the reviewers for their time and thorough evaluation of the manuscript. We believe that we have addressed all the raised comments which have substantially improved the quality of the manuscript.

Please see our detailed responses below

Reviewer 2

This is an interesting article, that provides new information regarding vaccine hesitancy among pregnant & lactating women in Jordan. I have some small comments to the authors.

Thank you for your comments.

Abstract: regarding the background, the description corresponds to the aims or objectives of the study, rather than the background information. More information is needed regarding vaccination rate in pregnant women, or why they are at specific risks if vaccinated during pregnancy. Also, it is important to know if at the moment of the study, pregnant women were or not getting the covid vaccine in Jordan.

The following was added to the abstract: “Pregnant women are at higher risk of developing severe COVID-19 symptoms. Therefore, booster dose against COVID-19 was recommended for this special population in Jordan. However, vaccine hesitancy/refusal remains the main obstacle of providing immunity against the spread of COVID-19.”

With respect to this comment in the abstract: “High perceived dangerousness of COVID-19 was significantly associated with better practice”, the better practice was actually getting the vaccine or being willing to receive it? It is not the same. Please clarify.

The practice as detailed in the manuscript evaluated participants adherence to the daily protective practices against COVID-19 including wearing face mask, preserving social distancing … etc. The intention toward receiving a booster dose was evaluated by a different question.

Key words: I suggest to review if these words are all considered within the MESH Subheadings (ie. Practice and intention probably are not).

Thank you for your comment. Intention is within MeSH subheadings, while practice is not, therefore, it was removed from keywords.

Introduction: It will be important to give information regarding the lack of clinical studies of COVID-19 vaccines that included pregnant and lactating women; most probably the lack of adequate clinical trials for these group of women contributed to vaccine hesitancy among this population and is an information that is relevant to consider.

The following was adding to the introduction “High hesitancy/refusal rates in this subgroup to receive COVID-19 vaccine could be attributed to the limited clinical studies that enrolled pregnant women. “

Likewise, it is important to have information regarding the Jordan vaccination program for COVID-19 (it included or not pregnant women).

The following was added to the introduction “Jordan has implemented a range of comprehensive and stringent preventive measures at multiple levels to effectively combat the transmission of COVID-19. However, as of 25th of May 2022, authorities in Jordan have implemented relaxed COVID-19 measures. The compulsory use of facemasks in both indoor and outdoor public areas is no longer mandatory. Furthermore, all restrictions on public gatherings, including those at places of worship and wedding venues, have been lifted. But all the preventable instructions were still recommended including wearing face mask, maintaining social distance of 3 feet in indoor setting, and hand hygiene. These decisions align with a decline in the number of COVID-19 cases. Regarding the receiving of booster dose of covid-19 it wasn’t obligated by low, however, it was recommended for the general population including pregnant and lactating woman.”

Comments on the Quality of English Language

Minor edits:

A total of 695 females enrolled in the study: were enrolled in the study (abstract)

This was corrected.

and safety concerns (5) Due (line 44), end of the sentence is missing.

A full stop was added.

with the majority had no chronic diseases, (line 124), please review the syntaxis.

This was corrected to” with the majority having no chronic diseases”

As shown in Table 4. The most (line 146); there is no need to end the phrase between Table 4 and “the most”…

This was corrected.

perceived dangerousness the infection with COVID-19 is (p-value <0.001) (line 187), please review the syntaxis.

The “is” was removed.

Reviewer 3 Report

COVID-19 vaccination booster dose: Acceptance, knowledge, practice, and intention among pregnant and lactating women

This manuscript describes the analysis of data taken from questionnaires given to women in Jordan attending gynaecology clinics (pregnant women, women planning to be pregnant, or lactating women). Those questionnaires sought to assess the level of acceptance for a COVID-19 booster dose in that population as the factors associated with that level of acceptance

Overall I found the manuscript to be interesting and potentially a useful contribution to the literature. There are, however, several useful revisions that I recommend before the manuscript is ready for publication. In general, the authors have failed to provide sufficient context for the situation in Jordan at the time the questionnaires were administered, rendering the results less useful to public health authorities and vaccine decision-making bodies. There are several results that do not have sufficient detail in the Methods section, so it is difficult to interpret those results. There are also some occasions where the authors need to modify their interpretation of those results in the Discussion section.

I’ll go through a line-by-line list of suggested revisions for the authors for each section, but here I’ll summarise the main revisions that need to be considered by the authors:

  • The authors need to clarify which booster dose they are referring to in their manuscript. Is this the first booster dose after completion of the primary vaccination series? Is it the second or third booster dose after the primary vaccination series? Is it any booster dose?

  • The authors need to add more context to the Introduction section to give their results more meaning. For example, what were the public health recommendations/obligations for mitigating the spread of SARS-CoV-2 at the time? When the authors mention social distancing, it isn’t clear if the authors mean trying to keep a certain distance from others in indoor spaces or if that instead refers to lockdowns and confinements at home. Understanding what was recommended or legally obliged at the time would help. Another major thing to clarify was how COVID-19 vaccines were available at the time in Jordan - were all vaccines freely available to the population or were some costs needed to be borne by members of the population to be vaccinated (line 264 in the Discussion mentions the affordability of vaccines, but if COVID-19 vaccines were free to the population then this comment in the Discussion is moot)? What were the specific recommendations for women included in this study?

  • The final major revision that I’ll suggest here, before going through a line-by-line analysis of the manuscript, is that the authors need to add many more descriptions of the questionnaires and questions used in the study. Several different scales have been used to quantify participants’ understanding of certain issues but none of those scales have been defined by the authors. I counted a 5-point scale for perceived “dangerousness” of COVID-19; a 10-point scale for vaccine reactions and side effects; a 22-point scale for knowledge of COVID-19; and a 25-point scale for practices to mitigate viral transmission.

COMMENTS SPECIFIC TO THE INTRODUCTION

The authors have clearly stated the research question in their Introduction section and the main objective of the study is clear. 

Line 40-41: “...vaccination remains superior in controlling the outbreak” should be reworded to something like “...preventative vaccination has been an extremely effective measure in reducing the incidence of COVID-19 and its complications, notably serious disease and disease-attributable mortality”.

Line 46: “...it is essential to implement booster strategies” should be reworded to something like “booster vaccination strategies have been implemented in several regions”.

Line 57: The phrase “An even higher rate of hesitancy…” when presenting the results from Italy by del Giudice et al. implies that the data on the preceding line also show vaccine hesitancy. I think this interpretation of the 42.9% vaccination coverage in Iran is a step too-far, and that we can’t assume that the unvaccinated pregnant women in the Iranian study are all vaccine-hesitant. Line 57 could be re-written as “A high rate of vaccine hesitancy…” to avoid this.

Line 60: The authors need to be specific when they say that del Giudice et al. showed “a lack of knowledge”...a lack of knowledge about what, exactly? Vaccine effectiveness, vaccine safety, vaccine availability, eligibility for vaccination, etc.?

The Introduction lacks context for the non-Jordanian reader or anyone not familiar with the pandemic mitigation strategies in the country. The authors need to inform the reader about the following:

  • What COVID-19 vaccination recommendations were in place for pregnant women, women planning to be pregnant and lactating women at the time of the study;

  • What obligations for vaccination were in place for the study population, if any (for example, were unvaccinated-but-eligible individuals breaking any laws by not getting vaccinated, or were they denied access to public spaces like restaurants and cafés, or were they denied access to the workspace, or any other restrictions)

  • What social distancing measures were recommended or obligated in Jordan at the time of the study, as the authors refer to social distancing without any details. This could mean recommending individuals maintain a distance between others when indoors, or it could be as strict as a legally-enforced lockdown or confinement at home. It could also refer to school closures, requirements to work from home rather than attend the office. The authors may not have been so detailed on their questionnaire, but the reader still needs to know what was happening in Jordan at the time so that they understand what the participants meant in their responses when asked about social distancing

For the issues described above, I will refer back to them when reviewing specific comments connected to the authors’ work in other sections of the manuscript.

COMMENTS SPECIFIC TO THE METHODS

Line 77: “our study included 695 participants” should be included in the Results section, not the Methods section

Line 88 (and repeated throughout the manuscript): “…knowing someone who died from COVID-19” should be re-written as the participant can never be entirely sure that their contact died as a result of COVID-19 infection. It should be something along the lines of “reported knowing someone who died due to COVID-19”. The authors should be presenting what was reported by the participants in the study, and doing so in this way avoids the potential debates over which contacts died due to COVID-19 and which died whilst suffering from but not because of COVID-19.

Lines 112 and 113: “These plots indicated that the data were not normally distributed” belongs in the Results section. The authors should state that the normality of variables was assessed with Q-Q plots, and that nonparametric analyses were conducted on any variables that were not normally distributed. It can then be reported in the Results section which variables were normally distributed and which were not. 

Lines 109 to 121: The authors need to clarify here that they are not referring to an analysis of the pilot study with 40 participants, but the analysis of the main study cohort. Upon first reading it was unclear if the authors were talking about a supplementary analysis of the pilot study or the full analysis dataset.

Lines 119 to 121: “The independent variables…in the bivariate analysis” should be re-written as “The independent variables in the three regression models were defined as the variables…”

One major issue in the Methods section is the missing descriptions of the various scales used in the questionnaire. I counted a 5-point scale for perceived “dangerousness” of COVID-19; a 10-point scale for vaccine reactions and side effects; a 22-point scale for knowledge of COVID-19; and a 25-point scale for practices to mitigate viral transmission. The authors need to mention in this section what those scales are (i.e. were they made by the authors for this study, or were they taken from the literature?) and what each extreme value in the scale means (i.e. in the perceived “dangerousness” scale, a score of 1 refers to XYZ and a score of 5 refers to XYZ. This should be clear for the reader for each scale used.

COMMENTS SPECIFIC TO THE RESULTS

It is noted that the authors managed to recruit 695 women to their study, but they have not specified how many women were offered the chance to participate. Are those data available?

Table 1 is interesting but it would be more useful to the reader if those data could be compared to some population data, particularly the population of women attending gynaecology clinics for the same reasons that the participants in the study attended those clinics. This way the reader can be assured that the cohort is not biassed demographically.

Line 129: “Most of the sample (77.7%) knew someone who had died due to COVID-19…” should instead read “Most of the sample (77,7%) reported to have known someone…”

The data on what side effects were reported from the participants. It would be helpful, although not essential, if at some point in the manuscript the authors could compare these reports to the clinical trials for the vaccines mentioned to see if the reported rates of side effects mirror the reports from the initial trials.

Line 146: “As shown in Table 4” is a fragment of a sentence. This needs to be edited as it makes no sense on its own.

Table 4 needs to be clarified for the reader as the reader will not necessarily know what the correct answer was for each question. An additional column could be added to the table to show what response was expected by the authors, with the next column then showing how many participants (and the percentage of the total) got that correct response. There’s no need to have the percentage of incorrect responses as it’s the complement of the percentage of correct responses, unless any responses were missing.

The authors also need to explain why the option for medical herb consumption was added to the table and to cite any studies that show that medical herbs were beneficial for managing COVID-19 symptoms or infection. At the moment there’s nothing in the manuscript that supports this claim and it needs to be either clarified or removed.

In Table 8 the participants responded to how often they wore face masks, but the authors need to comment in the Introduction section what was recommended/obliged during the study period in Jordan. If no recommendation or obligation was in place for face masks then 58.1% of people wearing masks would be impressive, but it would be less impressive if indeed mask wearing was obligatory at the time.

Line 176 (and mentioned several times in the manuscript): the concept of “dangerousness” needs clarification. The question posed to participants appears to have been “How dangerous is in the infection with COVID-19” but from both a clinical and public health point of view this is too vague. Are we talking about perceived risk of infection, the perceived risk of symptoms, of complications, of hospitalisation, of impact at home, of impact on personal finances, of mortality? The authors need to reconsider how this question is treated in the manuscript.

Lines 188-189: The authors need to specify the value of alpha that determined if a result was statistically significant or not. 

The authors repeatedly refer to a “high/low knowledge group, but they have not defined what score on the knowledge scale determined what the high knowledge group was. The same is true for the “high/low practice group” 

Line 216: “Participant” should read “Participants”

Line 227: The authors should define what they mean by both “vaccine hesitancy” and “vaccine refusal”. Are they using the responses to the question about uptake of the booster and assigning participants who responded “Maybe” as hesitant, and “No” and refusers?

COMMENTS SPECIFIC TO THE DISCUSSION AND CONCLUSION

Line 265-266: “...and can afford vaccines more easily…” implies that COVID-19 vaccines were not free to all members of the eligible population. The authors need to add some context about Jordan’s vaccine recommendations/obligations to the Introduction section, including if the vaccines were reimbursed (and it what degree) for the population

Line 280: “Based on these results…” which results? Those of reference 29 mentioned in the previous sentence or those of the manuscript? Please clarify.

Line 280: “It is critical…” should be re-written to avoid using the word “critical”, to use more neutral language.

Line 286: “Perception of the hazards connected to COVID-19…” should be re-written as the authors have used the words “dangerous” or “dangerousness” throughout the manuscript. I’ve suggested that those terms are reconsidered, and the new terms should also be used on line 286 so that the language is consistent throughout.

Line 288: “...like as….” should read “...such as…”

Line 288: “...to guard themselves against potential infection…” should read “...to reduce the probability of infection…”

Line 298: “dosage” should read “dose”

Line 302-303: “...potentially devastating symptoms…” should be re-written to use more neutral language.

Line 312: “...lack of research…” should read “...a lack of research”

Line 318-319: the authors can remove to phrase “as it is vital to safeguard both the mother’s health and the wellbeing of the growing fetus” 

Line 320: Change “worries” to “concerns”

Line 327: Remove the word “future”

COMMENTS SPECIFIC TO THE ABSTRACT

As written above, the authors should reconsider using the words "dangerous" and "dangerousness" and should instead use more specific language

COVID-19 vaccination booster dose: Acceptance, knowledge, practice, and intention among pregnant and lactating women

The standard of English used in this manuscript is very high and I do not recommend any separate English editing. I have made some minor suggestions to the authors in terms of grammar and use of non-neutral words, but nothing more is needed on that front.

Round 2

Reviewer 1 Report

Minor editing of English language required.

Author Response

Please find the responses attached. 

Reviewer 3 Report

I thank the authors for their careful consideration of my comments on the previous version of their manuscript.

They've added everything that I requested and I feel that the manuscript has been clarified to avoid any confusion for the reader. I'm satisfied that after the last stages of proof-reading that this manuscript is ready for publication.

I've noticed the occasional missing word in the manuscript that needs to be addressed before publication. An example is line 42 ("notably serious disease and disease-attributable **mortality**"). A proof-reader needs to highlight these instances so that they can be corrected before publication. But there's no need for major English language editing at all, the authors clearly have a very high level in that language.

Author Response

Response

We have now thoroughly proofread the manuscript, making further minor changes throughout.

The authors would like to thank the reviewer for your time and thorough evaluation of the manuscript. 

Round 3

Reviewer 1 Report

Thank you the authors for your time in consulting a statistician and responding my suggestions. The manuscript is now much improved, especially the results part. 

Minor editing of English language required